# How Does N Mineral Fertilizer Influence the Crop Residue N Credit?

**Risely Ferraz-Almeida [1],\* , Natália Lopes da Silva [2] and Beno Wendling [2]**

[1]  Departamento de Ciência do Solo, Universidade de São Paulo, Escola Superior "Luiz de Queiroz", Piracicaba, São Paulo 01246-903, Brazil

[2]  Instituto de Ciências Agrárias, Universidade Federal de Uberlândia, Uberlândia, Minas Gerais 38400, Brazil; natalialopesagro@gmail.com (N.L.d.S.); beno.iciag@gmail.com (B.W.)

\*  Correspondence: rizely@gmail.com; Tel.: +55-1934294100

**Abstract:** In no-tillage systems, there is an accumulation of crop residues (CR), which plays an essential role in the availability of soil-N. A study was set up to provide information regarding the N credit and the influence of N mineral fertilizer. There was the addition of a similar rate of residue (10 Mg ha$^{-1}$; sugarcane, soybean, and brachiaria) and N mineral fertilizer (urea; 120 kg N ha$^{-1}$) in loam soil. After the stabilization of biological activity (73 days), soil and remaining residues were collected, and C and N monitored. The results showed that the N credit was positive with the application of soybean, sugarcane, and brachiaria. There was a positive balance of the soybean N credit in soil with a reduction from 2.49 to 0.90 g kg$^{-1}$ of N in remaining residue, and a direct increase of 90% of soil-N. There is no need of N fertilizer to potentialize the soybean N credit, but it is required to potentialize N credit of brachiaria and sugarcane. The urea demonstrated to be an excellent enhancer of brachiaria N credit, but it was not adequate for sugarcane residues. Based on our result, the accumulation and incorporation of CR can be considered as N credit with a positive contribution in soil-N.

**Keywords:** sugarcane; soybean; trash; soil N; waste management

## 1. Introduction

In no-tillage systems, there is an accumulation of crop residues that drives the decomposition kinetics and impact the dynamic of nutrients in the soil. The accumulation of crop residues also increases the soil quality, impacting the aggregation, water retention, and cation exchange capacity in soil with a direct effect on crop production and the diversity of fauna and flora [1–3]. In the Mediterranean climate, Sapkota et al. [4] showed an increase of soil quality in a no-tillage system during 15 years, represented by a higher content of organic matter and the abundance and diversity of micro-arthropods. In a tropical climate, there is a rapid decomposition of residues in soil due to the adequate conditions of temperature and moisture for biological activities [2,5]. The accumulation of residue with inadequate management can also present adverse effects on soil quality. Sánchez-Rodríguez et al. [6] showed that crop residues exacerbate negatively impacted on soil quality indicators (e.g., redox potential and greenhouse gas emissions) and promoted an adequate condition for inoculation of plant diseases [7].

The dynamic of nitrogen (N) is affected by the accumulation of residues in soil because the soil N reserves are mainly found as organic N [8,9]. On the soil surface, about 90% of the N is organically combined [8]. Li et al. [10] showed that organic N in the soil could be classified according to availability (labile pool and stable pool) and chemical components (i.e., hydrolyzable and non-hydrolyzable, ammonia, amino acid, and amino sugar); the amino acid represents the highest N organic content in the soil, followed by non-hydrolyzable, ammonia, and amino sugar. The N in residue, when mineralized by

soil microorganisms, is converted into the mineral fractions (nitrate, $NO_3^-$; and ammonium, $NH_4^+$) [11]. $NO_3^-$ presents high mobility in the soil surface due to the low interaction and the predominance of negative charge in tropical soils. In comparison, $NH_4^+$ presents a higher use efficiency due to the low mobility in the soil surface. In the net N in the soil, there are N losses by nitrate leaching [12], nitrous oxide emissions [13,14], and ammonia volatilization [15].

The use of legume cover crops has been presented as an alternative to enrich soil N by accumulating and decomposing the remaining residue. The term "N credit" has been associated with soil N derived from the cultivation of soybean (*Glycine max* L.) in succession systems of soybean and corn (*Zea mays* L). The efficiency of N credit is variable due to the direct influence of biological activity in N mineralization. Ferraz-Almeida et al., Almeida et al., and Mikhael et al. [2,16,17] showed that the rate of residue mineralization in soil depends on temperature, soil moisture, and management of crop residues (i.e., rate and quality/source). Bundy et al. [18] showed that, in sandy soil, the remaining soybean residue provided a low quantity of N to subsequent crops due to N losses by leaching, but there was an increase in corn yield and N uptake in the subsequent crop in a silt loam soil. Tenelli et al. [19] noticed that the use of sunn hemp (*Crotalaria spectabilis*) as a cover crop promoted sugarcane (*Saccharum officinarum*) yield in subsequent years. However, there was no effect on the content of soil N. Possibly, the N credit of legumes (i.e., soybean, sunn hemp, bean (*Phaseolus vulgaris*), and lentil (*Lens culinaris*)), not only soybean, is also associated with the quality/quantify of N mineral fertilizer in subsequent years. Sapkota et al. [4] showed that the application of *Vicia villosa* (legume), known as the hairy vetch in Europe/Western Asia, can be considered as N organic source, and its effect is enhanced with a moderate N rate.

Even with positive benefits from crop residues in soil N dynamic, the application of mineral fertilizer is commonly performed in tropical conditions after legumes planting without counting the N credit. For example, the rate of 120 kg ha$^{-1}$ of N (mineral N fertilizers) is commonly performed in the production system of cereals in tropical conditions [20]. The mineral N fertilizers when applied in soil present a low N use efficiency in the production of grains (average of 50%) [21,22], due to soil N losses by ammonia volatilization [15], nitrous oxide emission [14], or nitrate leaching [23]. Fontoura et al. [24] showed that a reduction from 30–170 to 30–150 kg ha$^{-1}$ of N is required for the production of wheat (*Triticum aestivum* L.) when cultivated after soybean, while the production of barley (*Hordeum vulgare* L.) requires a reduction from 30–130 to 30–120 kg ha$^{-1}$. Therefore, it is necessary to consider the N derived from residue to an adequate balance of nitrogen fertilization in successive crops. Studies that demonstrate the legume N credit and influence of N in subsequent crop can contribute to decrease the use of mineral N and improve N credit supplied by the legumes. The reduction of conventional N fertilizers by organic N source derived by legume residues seems reasonable. If successful, the association of N sources (organic and mineral), isolated or associated, will achieve the principles of a circular economy with an adequate balance of society, economy, and the environment [25,26].

This study was set up to provide information regarding the N credit of crop residues as a corresponding N source associated with N mineral fertilizer. We hypothesized that crop residues (soybean, brachiaria, and sugarcane) could be an alternative to increase the N content in the soil. The goal here was to monitor inputs of N by applying crop residues and N mineral fertilizers, and its impacts on the content of N in soil and remaining residues.

## 2. Materials and Methods

### 2.1. Experimental Characterization

A study was run with the additions of residues (soybean, brachiaria, and sugarcane) and N mineral fertilizer in the soil, isolated or associated, using three replications in a completely randomized block design. Soil without inputs of residues or N mineral fertilizer was monitored as a control (Figure 1).

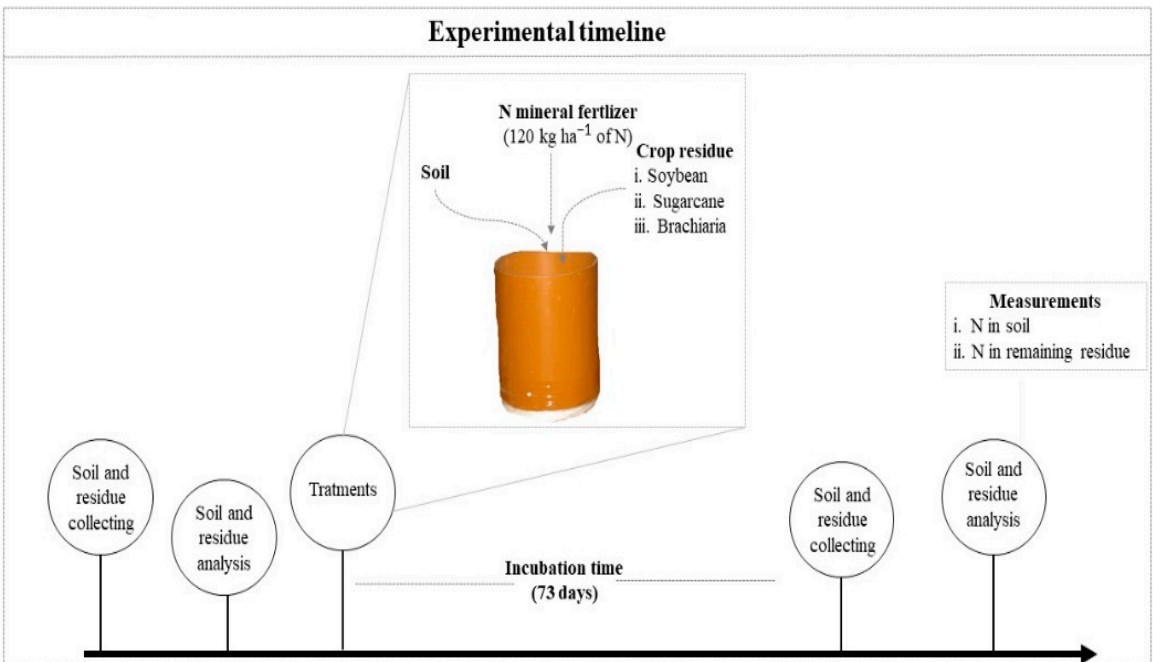

**Figure 1.** Experimental timeline of events with the additions of crop residues (soybean, sugarcane, and brachiaria) and nitrogen mineral fertilizer (N).

Soil and residues were collected from areas cultivated with soybean, brachiaria, and sugarcane, located in the region of Uberlândia and Uberaba (latitude 19°13′00.22″S and longitude 48°08′24.80″ W; 900 m). The soil was collected in the 0.0–0.2 m layer, and residues were removed from the soil sample. Soil and residues were chemically characterized [27]. Soil presented the following contents of phosphorus (2.47 mg $dm^{-3}$; Mehlich I), potassium (230.00 $cmol_c$ $dm^{-3}$), calcium (2.20 $cmol_c$ $dm^{-3}$), and magnesium (0.56 $cmol_c$ $dm^{-3}$), with a soil texture classified as loam, and respective contents of clay, silt, and sand: 230, 140, and 630 g $kg^{-1}$ (hydrometer method [28]). The soil was classified as a Latossolo Amarelo [29], corresponding to an Oxisol [30]. The total carbon (C) and N in residues were 120.00 and 1.44 g $kg^{-1}$ (sugarcane); 122.40 and 1.68 g $kg^{-1}$ (Brachiaria); and 142.12 and 2.49 g $kg^{-1}$ (soybean), respectively. The soil C and N were monitored by the acidified dichromate method [31] and the Kjeldahl method [27], respectively.

### 2.2. Additions of Soil and Residue

A soil sample of 700 g (size < 2 mm) was added in a column (volume: 1298.8 $cm^3$; height: 13 cm; diameter: 10.5 cm) and moistened at 60% of field capacity. A rate of 10 Mg $ha^{-1}$ (17 g of residue $pot^{-1}$) of the residues of soybean, sugarcane, and brachiaria was incorporated and homogenized in soil surface (at the first 5 cm) to promote the contact of soil and residue. We used this similar residue rate to understand the effect of residue quality in soil N content. In the field, the accumulation average of sugarcane, soybean, and brachiaria, respectively, range between 15 and 20 [32], 2 and 4 [33], and 5 and 2 Mg $ha^{-1}$ [34]. A rate of 120 kg $ha^{-1}$ of N mineral fertilizer (urea: 45% N) was incorporated and homogenized in soil (at the first 5 cm). The N rate was based on an average of current N recommendation for crop productions (i.e., corn, barley, and wheat) in tropical conditions [20,24].

The column was fixed on a Styrofoam-base to prevent the water loss, keeping the soil at 60% of field capacity, which was weighted continuously with the water addition when requested. This study was developed in the laboratory due to the controlled conditions, as demonstrated by Almeida et al. (2016) and Guo et al. (2015) [17,35]. The setup was left in an open environment (temperature, 25 °C) for 73 days. The time was based on previous studies with the goals to monitor the N in soil and remaining residues after the stabilization of biological activity in soil [16,17,36].

### 2.3. Measurements and Statistical Analysis

After the stabilization of biological activity (73 days), soil and remaining residues were collected and sieved (2-mm mesh). Soil and remaining residues were physically separated, and the contents of N were monitored using the Kjeldahl method [27]. The balances of N in soil and residues were calculated according to the difference of N contents at 73 days and the original N content (without N additions on soil).

The data were studied using descriptive statistics (average, standard deviation, minimum, maximum, and median values). The assumptions of residues normality and homogeneity of variance were tested using the Shapiro–Wilk test and the Bartlett test, respectively ($p \leq 0.05$). The data were submitted to the t-test to compare the two averages (average of treatments and control; $p \leq 0.05$) and the LSD-test (the Fisher test) to compare more than two averages (average of residues; $p \leq 0.05$).

Graphical algorithmic representation was used to demonstrate the inputs (crop residues and N mineral fertilizer) and outputs (N in the soil, N losses, and N in residue remaining). The positive outputs were correlated using the Pearson correlation ($p \leq 0.05$). All statistical analyses and graphical representations were performed using R (version 4.0.0; R Foundation for Statistical Computing, Vienna, Austria) and Python (version 3.8.3; Python Software Foundation, Wilmington, DE, USA).

## 3. Results

### 3.1. Total Soil N

With the general-average, the application of crop residues increased 85% of N in the soil, followed by an increase of 60% with the addition of N mineral fertilizer. The associated applications of both (crop residues and N mineral fertilizer) promoted higher N inputs in soil with an increase of 90% ($p < 0.05$), and, therefore, indicating an accumulated and positive effect of both N sources in soil N content (Figure 2A).

The highest soil N accumulation was found with soybean with a general average of 0.86 g kg$^{-1}$, followed by the application of brachiaria and sugarcane (Table 1). The associations of N mineral fertilizer and residues of soybean presented the highest soil N input with an average of 1.3 g kg$^{-1}$, but without a difference, if compared with the application of soybean residue (isolated; average: 1.1 g kg$^{-1}$) (Figure 2B). In addition, there was no significant difference in the application of sugarcane residue, and a small difference of 0.2 g kg$^{-1}$ between the average of residue and residue + N (Figure 2C). These results show an N input efficient in soil from soybean residue and a possible N loss in systems with N mineral fertilizer, mainly when associated with sugarcane residue. On the other hand, there was a significant effect of brachiaria residues and N mineral fertilizer, representing an increase of 42% compared to the application of residue isolated ($p < 0.05$) (Figure 2D).

**Table 1.** Descriptive statistics (average, median, standard error (SE), minimum, and maximum values) of N in soil and remaining residue with the application of crop residues (soybean, sugarcane, and brachiaria) and nitrogen mineral fertilizer.

| | Soybean | Sugarcane | Brachiaria | Average | Soybean | Sugarcane | Brachiaria | Average |
|---|---|---|---|---|---|---|---|---|
| | Total Soil N (g kg$^{-1}$) | | | | Remaining Residue N (g kg$^{-1}$) | | | |
| Average | 0.86 | 0.57 | 0.67 | 0.66 | 0.89 | 0.62 | 0.73 | 0.70 |
| Median | 0.84 | 0.56 | 0.56 | 0.62 | 0.90 | 0.65 | 0.85 | 0.67 |
| SE | 0.20 | 0.13 | 0.16 | 0.08 | 0.10 | 0.08 | 0.10 | 0.05 |
| Max | 1.54 | 1.12 | 1.33 | 1.54 | 1.15 | 0.82 | 1.00 | 1.15 |
| Min | 0.07 | 0.07 | 0.07 | 0.07 | 0.60 | 0.35 | 0.40 | 0.35 |

There were 3 and 12 observations for each residue and the general average, respectively. The treatments with an application of N mineral fertilizer are included in the average of residues.

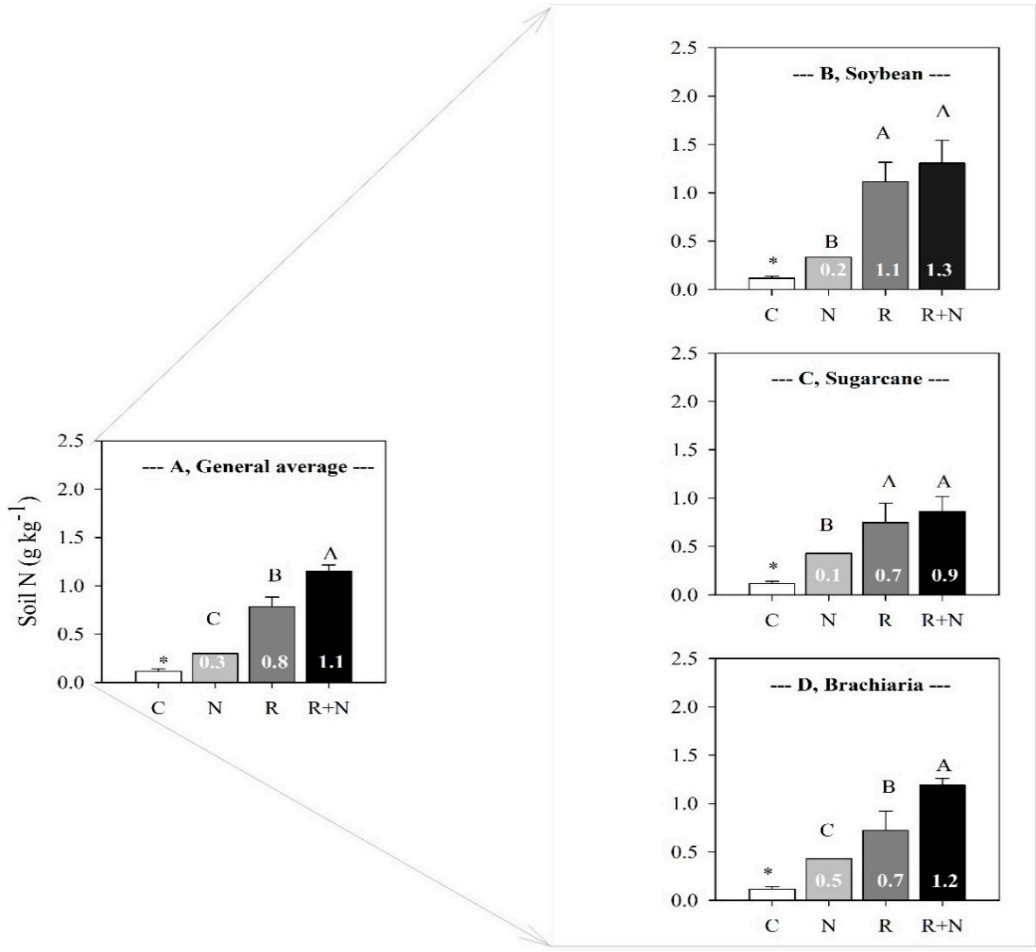

Applications of residue (R) and nitrogen (N)

**Figure 2.** (**A–D**) Soil N after application of crop residues (R; soybean, sugarcane, and brachiaria) and nitrogen mineral fertilizer (N). C represents the control treatment. The soil N in control was similar in all treatments (0.12 g kg$^{-1}$). Average of N, R, and R+N were tested by the LSD test ($p \leq 0.05$), and the general average of treatments (N, R, and R+N) and control were tested by the $t$-test ($p \leq 0.05$); uppercase letters and asterisks, respectively, represented these significant results.

The balance of soil N revealed that the associations of crop residue and N mineral fertilizer increased the N content in the soil, ranging from 84% to 90%, considered 5% higher than the application of isolated residue (Table 2). The highest positive balance of N in soil was observed with the application of soybean residue with sequences: soybean > sugarcane = brachiaria (residue); and soybean > brachiaria > sugarcane (residue + N mineral fertilizer). These results demonstrated that the soybean residue is an excellent N credit to increase N in the soil, and there was no need for an association of soybean with N mineral fertilizer (Table 2).

**Table 2.** Balance of total N in soil and remaining residue after the application of crop residues (R; soybean, sugarcane, and brachiaria) and nitrogen mineral fertilizer (N).

| | Soybean | Sugarcane | Brachiaria |
|---|---|---|---|
| | **Soil N (g kg$^{-1}$)** | | |
| R | +1.11 ± 0.2 (+90%) | +0.75 ± 0.2 (+84%) | +0.72 ± 0.2 (+84%) |
| R+N | +1.31 ± 0.1 (+91%) | +0.86 ± 0.1 (+86%) | +1.19 ± 0.1 (+90%) |
| Initial-N | 0.12 | 0.12 | 0.12 |
| | **Remaining Residue N (g kg$^{-1}$)** | | |
| R | −1.69 ± 0.2 (−68%) | −0.86 ± 0.1 (−59%) | −1.11 ± 0.1 (−66%) |
| R+N | −1.51 ± 0.1 (−61%) | −0.77 ± 0.1 (−54%) | −0.78 ± 0.1 (−46%) |
| Initial-N | 2.49 | 1.44 | 1.68 |

In initial-soil N, there was no addition of residue and N mineral fertilizer. The N balance is the difference between N content in initial-soil/residue and that at 73 days; the results are represented in real values (outside the parenthesis) and percentage (balance: positive, +; and negative, −).

*3.2. Remaining Residue Total N*

With the general average, there was a decrease of 62% and 50% of N in the remaining residue, respectively with the applications of residue and residue + N. This result indicates that the association of residue + N promoted a higher N accumulation in the residue (up to 0.2 g kg$^{-1}$), if compared with the application of residue (Figure 3A). The highest N accumulation in the remaining residue was found with soybean, followed by brachiaria and sugarcane (Table 1). In all residues, there was a reduction of N in remaining residue with a negative balance ranging from −46% to −68% (Table 2).

As expected, the N reduction in the remaining residue was more expressive in soybean with a decrease of −61% (residue + N) and −68 % (residue), respectively, representing a reduction from 2.49 (N in initial-residue) to 1.0 and 0.8 g kg$^{-1}$ (Table 2 and Figure 3B). There was no difference of N in remaining residue with the application of soybean and sugarcane residue, isolated or associated with N, respectively, represented by a general-average of 0.9 and 0.6 g kg$^{-1}$ of N (Figures 3B and 2C). In contrast, brachiaria and brachiaria + N decreased −66% and −46% of N in remaining residue with an average from 1.7 g kg$^{-1}$ of N in initial residue to 0.6 and 0.9 g kg$^{-1}$, respectively (Figure 3D). Therefore, these results indicate that there was mineralization of N in the soil, and a low N immobilization when associated with N mineral fertilizer. We also expected to find this result with the application of sugarcane residue, but there was no clear evidence of N immobilization with a general average ranging from 0.6 to 0.7 g kg$^{-1}$ of N in the remaining residue (Figure 3).

With the general average of treatments, we observed a positive correlation between the N in soil and the N in remaining residue with an r of 0.99 ($p \leq 0.05$). The N inputs in system were represented by the additions of crop residues and N mineral fertilizer, while the outputs were separated into positive (N in the soil and N in residue remaining) and negative (N losses). In our study, we monitored the positive outputs with the general averages of 1.3 ± 0.4, 1.2 ± 0.3, and 0.9 ± 0.1 g kg$^{-1}$ (N in soil) and 1.0 ± 0.3, 0.9 ± 0.2, and 0.7 ± 0.1 g kg$^{-1}$ (N in residue remaining), respectively, with the application of soybean, brachiaria, and sugarcane (Figure 4).

**4. Discussion**

The positive effect of N input in soil with the application of crop residues also is showed in the literature [37,38], as well as the positive effect of N mineral fertilizer [39–41]. In our study, the application of crop residues increased 87% of the total N in the soil, while the N mineral fertilizer increased 61% of the total N in the soil, and the association of crop residue and N mineral fertilizer contributed to increasing 90% of the total N in the soil. The positive effect of the application of crop residues was expected because the N organic sources play essential roles in soil N cycling with the additional benefits in soil quality, i.e., an increase of organic matter, enzymatic activity (β-glycosidases and urease), soil porosity, and water available [42–45].

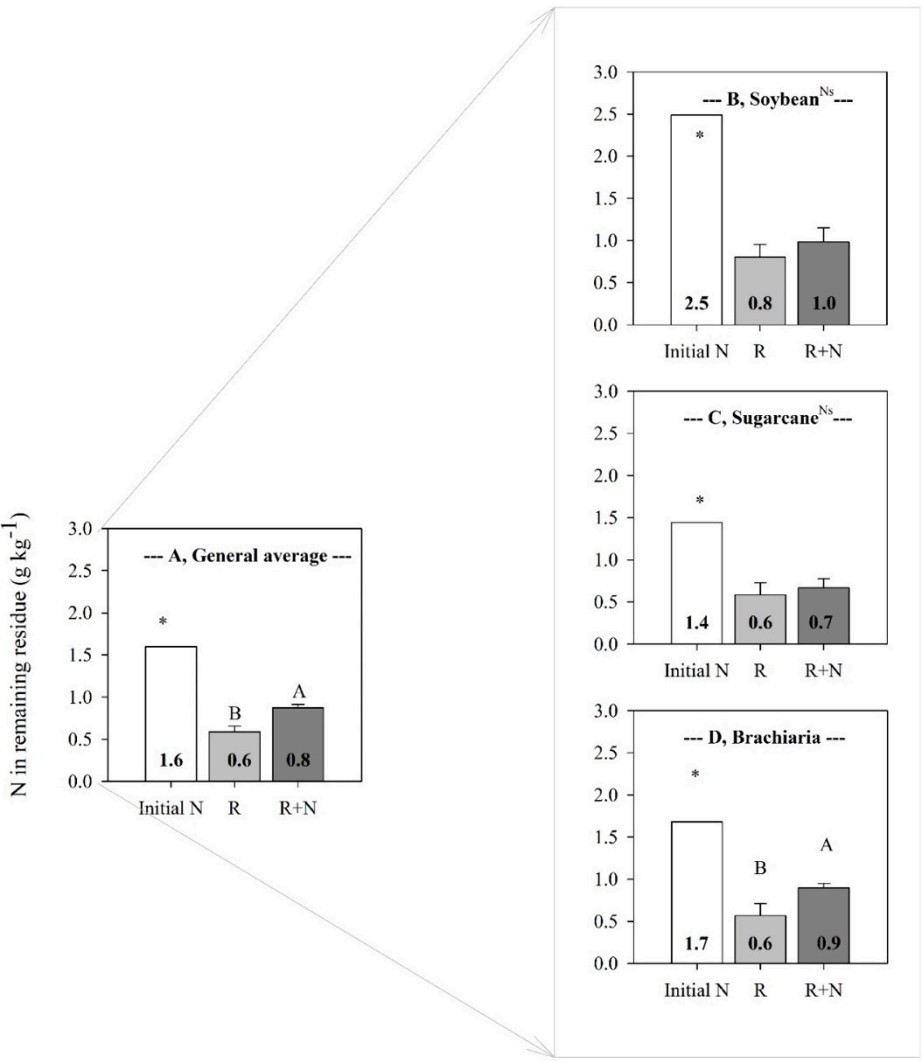

Applications of residue (R) and nitrogen (N)

**Figure 3.** (**A–D**) N in remaining residue after the application of crop residues (R; soybean, sugarcane, and brachiaria) and nitrogen mineral fertilizer (N). C represents control treatment. Average of R and R+N, and the average of treatments (R and R+N) and control were tested by the *t*-test ($p \leq 0.05$); the significant results are represented by uppercase letters and asterisks, respectively.

The soybean residue promoted the highest soil N accumulation indicating that soil microorganisms mineralized the total N content in the residue. In our study, there were no plants to check if the plants absorbed the N in the soil. In the field, Chioderoli et al., De Carvalho et al., and Torres et al. [46–48] showed that corn yield was positively impacted by the use of cover crops using leguminous (soybean or sunn hemp), explained by the increase of N content in the soil. The soybean residue increased by 90% and 92% the total soil N as a result of reduction of N in remaining residue by −68% and −56%, respectively, when applied residue isolated or associated with N mineral fertilizer. Increasing soil N with mineralization of soybean residues was presented by Almeida et al., Ferraz-Almeida et al., and Uchida et al. [16,17,49]. Maia et al. [37] showed that the application of beans and soybean residues and manure increased by 44% and 27% the total organic nitrogen in 0.0–0.1 and 0.1–0.2 m soil layers, respectively, compared to soil without N application. In our study, the highest decrease of N in remaining residue corroborated the N addition in soil by mineralization of soybean with a correlation of 99% (Pearson correlation). The soybean residue presented a low C:N rate and high N availability

(2.49 g kg$^{-1}$) to the activity of microorganisms in soil and N mineralization. The study of Rahman [50] with the application of residues (rice straw, rice root, cow dung, and poultry manure) noticed that the high amount of N and low C:N ratio in residue contributed to rapid microbial decomposition and increase of N in the soil. Therefore, our results and the results from the literature [16,17,49] indicate that soybean residue can be an N source due to high N credit in soil. The application of N mineral fertilizer did not increase the N credit of soybean residues with a small difference of 0.2 g kg$^{-1}$ between residue and residue + N. In the field, Blackmer [51] showed that the N credit of soybean residues is impacted by the time of residue decomposition, N content in residue, and rate of residue accumulation. Blumenthal et al. [52] also showed that the decisions made in soybean panting influenced the quantify of accumulated residue in soil (i.e., population density and root development). Possibly, in the field, studies using models, which can be based on an algorithm proposed (Figure 4), with different inputs of residues and climate conditions can help the decision of N application in the successive crop, turning the N credit of soybean as an active part of the plant N request.

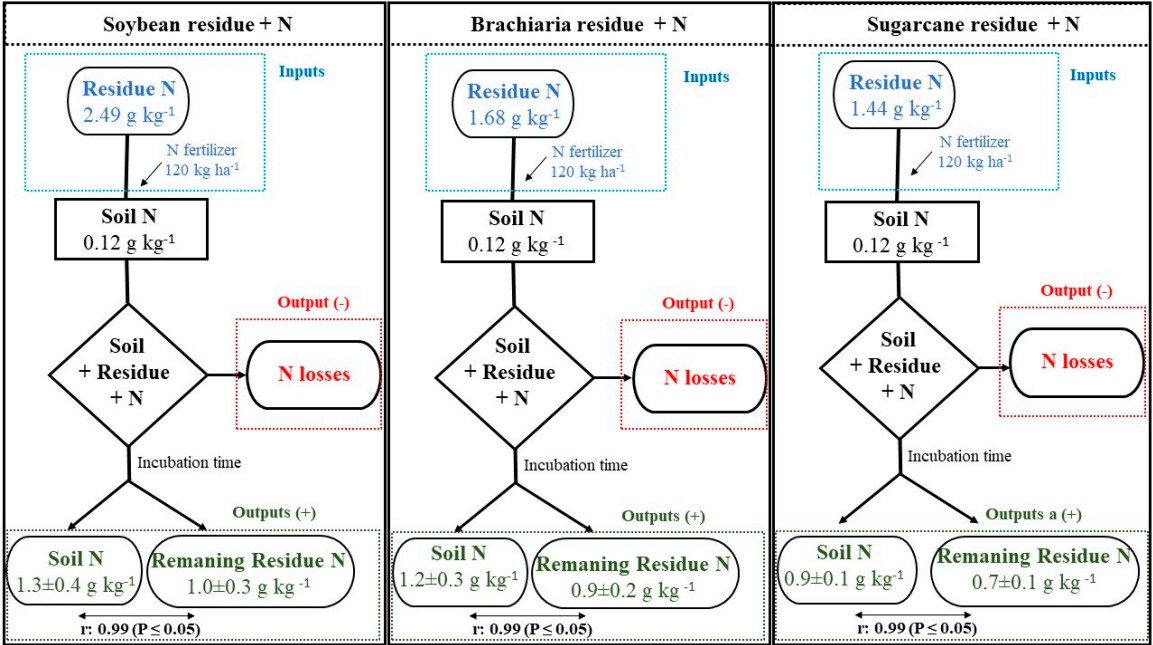

**Figure 4.** Graph algorithm with the inputs (crop residues and N mineral fertilizer) and outputs (soil N, N losses, and N in residue remaining) in the system. The positive outputs were correlated using the Pearson correlation ($p \leq 0.05$).

There was no difference between the application of sugarcane residues isolated or associated with N. The low N contribution with applications of sugarcane residue was also demonstrated by Meier et al. [38]. In tropical conditions, Fortes et al. [53] using $^{15}$N showed a total of 13% of N derived from sugarcane residues incorporated in the short term. Similar results were presented by Meier et al. [38] in Australia with a modest contribution of N from sugarcane residue. These results are explained by a high C:N ratio and low N content (1.44 g kg$^{-1}$) in residue that possibly promoted an immobilization of N in the residue. Wang et al. and Gunnarsson et al. [54,55] demonstrated that the high C:N in residues is associated with the content of cellulose, hemicellulose, and lignin. For residues with high C concentrations, e.g. sugarcane, microorganisms request several cycles and extra time to decompose it [50]. Ferreira et al. and Fortes et al. [53,56] described that the immobilization of N in sugarcane residue is a reserve of N, being a slow-release source in the long term.

Interestingly, we expected that the application of N would promote the increase of mineralization and content of N in the soil. However, in our study, we did not notice this result with a small difference of 0.2 g kg$^{-1}$ between residue and residue + N. Possibly, this result can be associated with the low

rate of residues tested, which was 50% lower than in the field. Besides, the application of N as a source of urea is characterized by the high losses of N through ammonia volatilization, mainly when surface-applied to soils [57–60]. Costa et al. [61] showed that N lost by application of N as urea can achieve 35% of applied N when associated with sugarcane residue. Another perspective can be the residue contamination with soil. The next studies will probably consider others methods to physical separation of residue and soil to avoid.

The addition of brachiaria residues was an intermediate N credit compared with the result of soybean and sugarcane residue. The intermediate brachiaria N credit was expected because the C:N ratio and N content (1.68 g kg$^{-1}$) in the residue were intermediate if compared to soybean and sugarcane residue. The application of N in brachiaria residue enhanced the N credit of brachiaria residues, causing an increase of 42% of soil N as a response to reducing 60% of N in the remaining residue. However, even with an increase of soil N, there was a low immobilization of N in remaining residue with N application, which reinforces our idea of residue contamination with soil. The significant contributions of brachiaria residues in soil N also are shown in the literature [2,62]. Torres et al. [36] showed an increase of soil N with the addition of brachiaria residues, considered a higher contribution than the residues of sorghum (84 kg ha$^{-1}$; *sorghum bicolor*), sunn hemp (118 kg ha$^{-1}$; *Crotalaria juncea*), black oat (29 kg ha$^{-1}$; *Avena strigosa Schreb*), and pigeon pea (51 kg ha$^{-1}$; Cajanus cajan (L.) Mill sp), but with a lower N contribution than millet (165 kg ha$^{-1}$; *Pennisetum glaucum*). The influence of brachiaria residue is associated with the volume of residues added in soil (surface and subsurface) [17]. The results of Torres et al. [36] were explained by the high residue rates. Mikhael et al. [2] also showed that N addition in the pasture area is a great strategy to increase the N stocks in soil and avoid soil depletion.

## 5. Conclusions

The N credit of crop residues was positive with the application of soybean, sugarcane, and brachiaria residue. There was a positive balance of the soybean N credit in soil with a reduction from 2.49 to 0.9 g kg$^{-1}$ of N in remaining residue, and a direct increase of 90% of soil-N. There is no need for N mineral fertilizer to potentialize the N credit with the incorporation of soybean residues. However, it is required with accumulation and establishment of brachiaria and sugarcane residues in soil. Urea demonstrated to be a great enhancer of N credit in brachiaria residue, but it is not adequate for sugarcane residues due to the N losses in the system, which are well-demonstrated in the literature [59,61]. Based on our result, the accumulation and incorporation of crop residues can be considered an N organic source with a positive contribution in soil N due to mineralization of organic matter.

**Author Contributions:** Conceptualization, R.F.-A., N.L.d.S., and B.W.; Methodology, R.F.-A., N.L.d.S., and B.W.; Formal Analysis, R.F.-A.; Writing—Original Draft Preparation, R.F.-A., N.L.d.S., and B.W.; and Writing—Review and Editing, R.F.-A., N.L.d.S., and B.W. All authors have read and agreed to the published version of the manuscript.

**Funding:** This research was funded by the Coordenação de Aperfeiçoamento de Pessoal de Nível Superior (CAPES; grant number, 88882.317567/2019-01).

**Acknowledgments:** Thanks are given to the Coordenação de Aperfeiçoamento de Pessoal de Nível Superior (CAPES) and the Universidade de Uberlândia, campus Uberândia.

**Conflicts of Interest:** The authors declare no conflict of interest.

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
