# Peer review of "How Does N Mineral Fertilizer Influence the Crop Residue N Credit?"

_nitrogen, doi:10.3390/nitrogen1020009_

Round 1

Reviewer 1 Report

Dear Authors,

I have studied your article and am of the opinion that it may be published in Nitrogen journal. However, it will be necessary to make some changes and edit the article. Because in its current form it contains some inaccuracies and potential mistakes.

Abstract

Lines 12 - 25: This part is brief and clear. I have no comment

Introduction

Lines 29 – 77: This part is brief and clear. I'm afraid you used the wrong citation system. This chapter clearly states that: References should be numbered in order of appearance and indicated by a numeral or numerals in square brackets, e.g., [1] or [2,3], or [4–6].

Line 64: You point out the positive effects of post-harvest residues (It is ok, but..). Would it not be objective to state the negatives of no-till systems? For example: increased hydrophobicity of the surface layer; survival of pathogens (mikromycetes); creation of a suitable habitat for rodent reproduction → damage to the harvest etc.  

Materials and Methods

Lines 79 – 128: I'm afraid again you used the wrong citation system. This chapter clearly states that: References should be numbered in order of appearance and indicated by a numeral or numerals in square brackets, e.g., [1] or [2,3], or [4–6].

Experimental characterization

Line 80: A description of the experiment is not fully realized the obvious. Would it be possible to add some scheme according to which the sampling was carried out?

Measurements and statistical analysis

Line 121:  Why did you use the LSD test? I assume you used the Fisher test, right?

Line 127: All analysis and graphical data processing was performed in software R?

Results 

Lines 130 – 198: I'm afraid again you used the wrong citation system. This chapter clearly states that: References should be numbered in order of appearance and indicated by a numeral or numerals in square brackets, e.g., [1] or [2,3], or [4–6]. The text is relatively difficult to read due to the citation system used.

I appreciate the detailed description of the results.

Dicsussion

Lines 200 - 273: I'm afraid again you used the wrong citation system. This chapter clearly states that: References should be numbered in order of appearance and indicated by a numeral or numerals in square brackets, e.g., [1] or [2,3], or [4–6].

Line 205: “…N organic sources played an important role in soil 204 N cycling with additional benefices in soil quality, i.e. increase of organic matter, enzymatic activity…”

Line 204: “…N organic sources played an important role in soil 204 N cycling with additional benefices in soil quality, i.e. increase of organic matter, enzymatic activity…For what enzymatic activities? Beware of this statement if you do not have the results of your own measurement of enzymatic activities. It has been shown, for example in the case of DHA that an increased content of N substances in the soil can have an inhibitory effect on this enzymatic activity.

Conclusion

Lines 275 - 285: This part is brief and clear. I have no comment.

Author Response

RESPONSE TO REVIEWER 1

 Editions in blue in Manuscript

Reviewer 1: Abstract: Lines 12 - 25: This part is brief and clear. I have no comment

Authors: Thanks. We glad to know it.

Reviewer 1: Introduction, Lines 29 – 77: This part is brief and clear. I'm afraid you used the wrong citation system. This chapter clearly states that: References should be numbered in order of appearance and indicated by a numeral or numerals in square brackets, e.g., [1] or [2,3], or [4–6].

Authors: Thanks. We checked all references in manuscript.

Reviewer 1: Line 64: You point out the positive effects of post-harvest residues (It is ok, but..). Would it not be objective to state the negatives of no-till systems? For example: increased hydrophobicity of the surface layer; survival of pathogens (mikromycetes); creation of a suitable habitat for rodent reproduction → damage to the harvest etc. 

Authors: Thanks. We agree with your point of view. We added information about the negative effect of exacerbate accumulation of crop residue in soil.

Reviewer 1: Materials and Methods. Lines 79 – 128: I'm afraid again you used the wrong citation system. This chapter clearly states that: References should be numbered in order of appearance and indicated by a numeral or numerals in square brackets, e.g., [1] or [2,3], or [4–6].

Authors: Thanks. We checked all references in manuscript.

Reviewer 1: Experimental characterization. Line 80: A description of the experiment is not fully realized the obvious. Would it be possible to add some scheme according to which the sampling was carried out?

Authors: Thanks. A Figure of treatments was added in Material and Methods to clarify the experimental characterization.

Reviewer 1: Measurements and statistical analysis. Line 121:  Why did you use the LSD test? I assume you used the Fisher test, right? Line 127: All analysis and graphical data processing was performed in software R?

Authors: Thanks. You are correct, we edited this part to make our text clear to reader.

Reviewer 1: Results . Lines 130 – 198: I'm afraid again you used the wrong citation system. This chapter clearly states that: References should be numbered in order of appearance and indicated by a numeral or numerals in square brackets, e.g., [1] or [2,3], or [4–6]. The text is relatively difficult to read due to the citation system used. I appreciate the detailed description of the results.

Authors: Thanks. We checked all references in manuscript. We are glad to know that appreciated the detailed description of the results.

Reviewer 1: Dicsussion. Lines 200 - 273: I'm afraid again you used the wrong citation system. This chapter clearly states that: References should be numbered in order of appearance and indicated by a numeral or numerals in square brackets, e.g., [1] or [2,3], or [4–6].

Authors: Thanks. We checked all references in manuscript.

Reviewer 1: Line 205: “…N organic sources played an important role in soil 204 N cycling with additional benefices in soil quality, i.e. increase of organic matter, enzymatic activity…” Line 204: “…N organic sources played an important role in soil 204 N cycling with additional benefices in soil quality, i.e. increase of organic matter, enzymatic activity…” For what enzymatic activities? Beware of this statement if you do not have the results of your own measurement of enzymatic activities. It has been shown, for example in the case of DHA that an increased content of N substances in the soil can have an inhibitory effect on this enzymatic activity.

Authors: Thanks. Perfect. We exemplify the β-glycosidases and urease. We noticed the increase of β-glicosidases and urease activities in previous studies with addition of residues in soil.

Reviewer 1: We checked all references in manuscript.

Authors: Thanks. We checked all references in manuscript.

Reviewer 1: Conclusion; Lines 275 - 285: This part is brief and clear. I have no comment.

Authors: Thanks. We are glad to know that you had no comments about our conclusion.

Reviewer 2 Report

The topic of the manuscript titled  ,,How Does N Mineral Fertilizer Influence the Crop Residue N Credit?’’ is interesting and results are valuable, but it should be corrected.

The Abstract is written according to the Journal guidelines. I have only one comment: 

  1. Line 16 - ,,After the stabilization of biological activity’’ Provide it in more detail. How many days?

In the Introduction section the authors well described the research background and current knowledge. However, I have some comments for this section:

  1. References must be provided as normal font (not superscript), e.g., [1] or [2,3], or [4–6]. See ,,Instructions for Authors’’ on the Journal website.
  2. Lines 49-52 – the references are needed for this definition.
  3. You must provide the name of the first author in some cases eg. Lines: 33 ,,In Mediterranean climate, Sapkota et al. [4] showed an increasing….’’

The same remark applies to sentences in Lines: 53, 55, 57, 61.

  1. See comment 1 in M&M section.

Materials and Methods section:

  1. Lines 103-105. ,,A rate of 120 kg ha-1 of N mineral fertilizer (urea: 45% N) was incorporated and homogenized also in soil surface (at the first 5 cm). The N rate was based in an average of current N recommendation for crop productions in tropical conditions’’.

The references are needed for the last sentence. Moreover, it is still not clear, why this rate was used.

An additional paragraph is needed in the Introduction section on the use of mineral nitrogen with post-harvest residues. What nitrogen doses are used? What does the nitrogen dose depend on? What doses are economically justified? Is this nitrogen dose included in the overall balance of nitrogen fertilization in successive crops and how?

Results section:

  1. Line 137 – Correct Figure 1. caption ,,Total soil N after the incubation of crop residues (R; soybean, sugarcane and brachiaria) and…’’
  2. Avoid giving the same values in the text and tables, e.g. Lines 143-144; 171-172; 196-198.

Discussion section:

  1. This section is well written.
  2. Lines 224, 242, 243, 246,  248, 255, 266271, 272 – provide the first author’s 
  3. Line 229 Correct the sentence ,,In field, Blackmer (1996) [44] showed…’’
  4. Line 239 Correct Figure 3. Caption ,,… using Pearson correlation (P ≤ 0.05). .’’
  5. References must be provided as normal font (not superscript).

Author contributions:

  1. Specify Author Contributions. See manuscript template.

General comments:

  1. Carefully check Instruction for Authors and manuscript template.
  2. Check the references citation throughout the manuscript.

Author Response

RESPONSE TO REVIEWER 2

Editions in green in Manuscript

Reviewer 1: The Abstract is written according to the Journal guidelines. I have only one comment: Line 16 - ,,After the stabilization of biological activity’’ Provide it in more detail. How many days?

Authors: Thanks. We glad to know that our abstract is well written. We provided more detail of how many day

Reviewer 1: In the Introduction section the authors well described the research background and current knowledge. However, I have some comments for this section:

Authors: Thanks. We glad to know that our abstract is well written.

Reviewer 1: References must be provided as normal font (not superscript), e.g., [1] or [2,3], or [4–6]. See ,,Instructions for Authors’’ on the Journal website.

Authors: Thanks. All references were edited according to the instructions for Authors’’.

Reviewer 1: Lines 49-52 – the references are needed for this definition.

You must provide the name of the first author in some cases eg. Lines: 33 ,,In Mediterranean climate, Sapkota et al. [4] showed an increasing….’’The same remark applies to sentences in Lines: 53, 55, 57, 61.

Authors: Thanks. All references were edited according your suggestions and the instructions for Authors’’.

Reviewer 1: See comment 1 in M&M section. Materials and Methods section: Lines 103-105. ,,A rate of 120 kg ha-1 of N mineral fertilizer (urea: 45% N) was incorporated and homogenized also in soil surface (at the first 5 cm). The N rate was based in an average of current N recommendation for crop productions in tropical conditions’’. The references are needed for the last sentence. Moreover, it is still not clear, why this rate was used.

Authors: Thanks. References were edited according your suggestions, and an explanation was added to explain the use of N.

Reviewer 1: An additional paragraph is needed in the Introduction section on the use of mineral nitrogen with post-harvest residues. What nitrogen doses are used? What does the nitrogen dose depend on? What doses are economically justified? Is this nitrogen dose included in the overall balance of nitrogen fertilization in successive crops and how?

Authors: Thanks. We added a paragraph in the Introduction with the information requested.

Reviewer 1: Results section: Line 137 – Correct Figure 1. caption ,,Total soil N after the incubation of crop residues (R; soybean, sugarcane and brachiaria) and…’’

Avoid giving the same values in the text and tables, e.g. Lines 143-144; 171-172; 196-198.

 Authors: Thanks. We added some part of the text to avoid giving the same values in the text and tables. We appreciate the suggestion to improve our writing.

Reviewer 1: Discussion section: This section is well written.; Lines 224, 242, 243, 246,  248, 255, 266271, 272 – provide the first author’s; Line 229 Correct the sentence ,,In field, Blackmer (1996) [44] showed…’’; Line 239 Correct Figure 3. Caption ,,… using Pearson correlation (P ≤ 0.05). .’’; References must be provided as normal font (not superscript).

Authors: Thanks. All suggestions were added in text.

Reviewer 1: Author contributions: Specify Author Contributions. See manuscript template.

Authors: Thanks. The author contributions were edited.
